# Ultrasound Cut-Off Values for Rectus Femoris for Detecting Sarcopenia in Patients with Nutritional Risk

**DOI:** 10.3390/nu16111552

**Published:** 2024-05-21

**Authors:** Daniel de Luis Roman, José Manuel García Almeida, Diego Bellido Guerrero, Germán Guzmán Rolo, Andrea Martín, David Primo Martín, Yaiza García-Delgado, Patricia Guirado-Peláez, Fiorella Palmas, Cristina Tejera Pérez, María García Olivares, María Maíz Jiménez, Irene Bretón Lesmes, Carlos Manuel Alzás Teomiro, Juan Manuel Guardia Baena, Laura A. Calles Romero, Inmaculada Prior-Sánchez, Pedro Pablo García-Luna, María González Pacheco, Miguel Ángel Martínez-Olmos, Blanca Alabadí, Valeria Alcántara-Aragón, Samara Palma Milla, Tomás Martín Folgueras, Andrea Micó García, Begoña Molina-Baena, Henry Rendón Barragán, Pablo Rodríguez de Vera Gómez, María Riestra Fernández, Ana Jiménez Portilla, Juan J. López-Gómez, Nuria Pérez Martín, Natalia Montero Madrid, Alba Zabalegui Eguinoa, Cristina Porca Fernández, María José Tapia Guerrero, Marta Ruiz Aguado, Cristina Velasco Gimeno, Aura D Herrera Martínez, María Novo Rodríguez, Natalia C. Iglesias Hernández, María de Damas Medina, Irene González Navarro, Francisco Javier Vílchez López, Antía Fernández-Pombo, Gabriel Olveira

**Affiliations:** 1Endocrinology and Nutrition Department, Clinical Universitary Hospital of Valladolid, 47003 Valladolid, Spain; dprimoma@saludcastillayleon.es (D.P.M.); jlopezgo@saludcastillayleon.es (J.J.L.-G.); 2Investigation Centre Endocrinology and Nutrition, Faculty of Medicine, University of Valladolid, 47003 Valladolid, Spain; 3Department of Endocrinology and Nutrition, Hospital Universitario Virgen de la Victoria, 29010 Málaga, Spain; jgarciaalmeida@gmail.com (J.M.G.A.); pguirado1991@gmail.com (P.G.-P.); nataly_montero@hotmail.com (N.M.M.); 4Department of Endocrinology and Nutrition, Complejo Hospitalario Universitario de Ferrol (CHUF), 15405 Ferrol, A Coruña, Spain; cristina.tejera.perez@sergas.es (C.T.P.); cris88_pf@hotmail.com (C.P.F.); 5Medical Department, Abbott Laboratories, 28050 Madrid, Spain; german.guzman1@abbott.com (G.G.R.); andrea.martin@abbott.com (A.M.); 6Department of Endocrinology and Nutrition, Complejo Hospitalario Universitario Insular-Materno Infantil, 35016 Gran Canaria, Spain; ygarciadelgado@gmail.com (Y.G.-D.); nuriapm@hotmail.com (N.P.M.); 7Institute of Biomedical Research in Malaga (IBIMA)-Bionand Platform, University of Malaga, 29590 Málaga, Spain; mery.garcia.96@gmail.com (M.G.O.); mjtapiague@gmail.com (M.J.T.G.); gabrielm.olveira.sspa@juntadeandalucia.es (G.O.); 8Endocrinology and Nutrition Department, Hospital Universitari Vall D’Hebron, 08036 Barcelona, Spain; fiorellaximena.palmas@vallhebron.cat (F.P.); alba.zabalegui@vallhebron.cat (A.Z.E.); 9Epigenomics in Endocrinology and Nutrition Group, Santiago Health Research Institute (IDIS), 34113 Santiago de Compostela, Spain; 10Endocrinology and Nutrition Department, Hospital Regional Universitario de Málaga, 29010 Málaga, Spain; 11Department of Medicine and Surgery, Malaga University, 29010 Malaga, Spain; 12Department of Endocrinology and Nutrition, University Hospital 12 de Octubre, 28041 Madrid, Spain; mariamaizj@gmail.com (M.M.J.); marta.ruiz84@gmail.com (M.R.A.); 13Endocrinology and Nutrition Unit, Hospital General Universitario Gregorio Marañón, 28007 Madrid, Spain; irenebreton@gmail.com (I.B.L.); cvelascog@salud.madrid.org (C.V.G.); 14Reina Sofía University Hospital, 14004 Córdoba, Spain; alzas94@hotmail.com (C.M.A.T.); aurita.dhm@gmail.com (A.D.H.M.); 15Service of Endocrinology and Nutrition, University Hospital Virgen de las Nieves, 41013 Granada, Spain; guardiabaena@gmail.com (J.M.G.B.); marianovor@hotmail.com (M.N.R.); 16Endocrinology and Nutrition Service, Basurto University Hospital, 48903 Bilbao, Spain; lauraaraceli.callesromero@osakidetza.eus (L.A.C.R.); nataliacovadonga.iglesiashernandez@osakidetza.eus (N.C.I.H.); 17Endocrinology Service, Complejo Hospitalario de Jaén, 23007 Jaén, Spain; priorsanchez@hotmail.com (I.P.-S.); mdedamasm@gmail.com (M.d.D.M.); 18Endocrinology and Nutrition Clinical Management Unit, Institute of Biomedicine of Seville (IBiS), Hospital Universitario Virgen del Rocío/CSIC/University of Seville, 41013 Seville, Spain; garcialunapp@yahoo.es (P.P.G.-L.); irenegonzalez1@gmail.com (I.G.N.); 19Endocrinology and Nutrition Department, Hospital Universitario Puerta del Mar, 11009 Cádiz, Spain; mariagonzaleznutricion@gmail.com (M.G.P.); franvilchez1977@gmail.com (F.J.V.L.); 20Biomedical Research and Innovation Institute of Cádiz (INiBICA), 11009 Cádiz, Spain; 21Division of Endocrinology and Nutrition, University Clinical Hospital of Santiago de Compostela, 15706 Santiago de Compostela, Spain; miguel.angel.martinez.olmos@sergas.es (M.Á.M.-O.); antiafpombo@gmail.com (A.F.-P.); 22Molecular Endocrinology Research Group, Health Research Institute of Santiago de Compostela-IDIS, 15706 Santiago de Compostela, Spain; 23CIBERObn, National Health Institute Carlos III, 28031 Madrid, Spain; 24Service of Endocrinology and Nutrition, Hospital Clínico Universitario of Valencia, 46010 Valencia, Spain; blanca.alabadi@gmail.com; 25INCLIVA Biomedical Research Institute, 46010 Valencia, Spain; 26Endocrinology and Nutrition Department, Marqués de Valdecilla University Hospital, 39008 Santander, Spain; v.alcantara.aragon@gmail.com; 27Clinical Nutrition and Dietetics Unit, Endocrinology and Nutrition Service, University Hospital of La Paz, 28046 Madrid, Spain; palmamillasamara@gmail.com; 28Endocrinology and Nutrition Service, Hospital Universitario de Canarias, 38320 Tenerife, Spain; tmf7312@hotmail.com; 29Endocrinology and Nutrition Service, La Fe University and Polytechnic Hospital, 46026 Valencia, Spain; andrea.mico@hotmail.com; 30Deparment of Endocrinology and Nutrition, Hospital Universitario de la Princesa, Instituto de Investigación Sanitaria Princesa (IIS-IP), 28006 Madrid, Spain; bmolinabaena@hotmail.com; 31Endocrinology and Nutrition, FEA Endocrinology and Nutrition, Navarra University Hospital, 31008 Pamplona, Spain; hereba17@gmail.com; 32Endocrinology and Nutrition Service, Virgen Macarena University Hospital, 41009 Seville, Spain; pablordevera@gmail.com; 33Endocrinology Service, Cabueñes University Hospital, 33394 Gijón, Spain; mriestra.fernandez@gmail.com; 34Health Research Institute of the Principality of Asturias (ISPA), 33011 Oviedo, Spain; 35Endocrinology and Nutrition Service, Consorci Hospital General Universitari de València, 46014 Valencia, Spain; anajipor@gmail.com; 36Department of Medicine and Dermatology, Faculty of Medicine, University of Malaga, 29010 Málaga, Spain; 37Professor Novoa Santos Foundation, 15405 A Coruña, Spain; 38i+12 Research Institute, 28041 Madrid, Spain; 39Maimonides Institute for Biomedical Research of Cordoba (IMIBIC), 14004 Córdoba, Spain; 40CIBER of Diabetes and Associated Metabolic Diseases, Carlos III Health Institute, 28029 Madrid, Spain

**Keywords:** sarcopenia, nutritional ultrasound, ultrasound cut-off values, rectus femoris, malnutrition, nutritional risk

## Abstract

Background: A nationwide, prospective, multicenter, cohort study (the Disease-Related caloric-protein malnutrition EChOgraphy (DRECO) study) was designed to assess the usefulness of ultrasound of the rectus femoris for detecting sarcopenia in hospitalized patients at risk of malnutrition and to define cut-off values of ultrasound measures. Methods: Patients at risk of malnutrition according to the Malnutrition Universal Screening Tool (MUST) underwent handgrip dynamometry, bioelectrical impedance analysis (BIA), a Timed Up and Go (TUG) test, and rectus femoris ultrasound studies. European Working Group on Sarcopenia in Older People (EWGSOP2) criteria were used to define categories of sarcopenia (at risk, probable, confirmed, severe). Receiver operating characteristic (ROC) and area under the curve (AUC) analyses were used to determine the optimal diagnostic sensitivity, specificity, and predictive values of cut-off points of the ultrasound measures for the detection of risk of sarcopenia and probable, confirmed, and severe sarcopenia. Results: A total of 1000 subjects were included and 991 of them (58.9% men, mean age 58.5 years) were evaluated. Risk of sarcopenia was detected in 9.6% patients, probable sarcopenia in 14%, confirmed sarcopenia in 9.7%, and severe sarcopenia in 3.9%, with significant differences in the distribution of groups between men and women (*p* < 0.0001). The cross-sectional area (CSA) of the rectus femoris showed a significantly positive correlation with body cell mass of BIA and handgrip strength, and a significant negative correlation with TUG. Cut-off values were similar within each category of sarcopenia, ranging between 2.40 cm^2^ and 3.66 cm^2^ for CSA, 32.57 mm and 40.21 mm for the *X*-axis, and 7.85 mm and 10.4 mm for the *Y*-axis. In general, these cut-off values showed high sensitivities, particularly for the categories of confirmed and severe sarcopenia, with male patients also showing better sensitivities than women. Conclusions: Sarcopenia in hospitalized patients at risk of malnutrition was high. Cut-off values for the better sensitivities and specificities of ultrasound measures of the rectus femoris are established. The use of ultrasound of the rectus femoris could be used for the prediction of sarcopenia and be useful to integrate nutritional study into real clinical practice.

## 1. Introduction

Sarcopenia has been classically defined as an age-related decline in skeletal muscle mass and function with adverse effects on quality of life and survival [1], but the European Working Group on Sarcopenia in Older People (EWGSOP2) has recently published an updated consensus definition that uses low muscle strength as the key characteristic for the condition rather than low muscle mass [2]. When low muscle strength, low muscle quantity/quality, and low physical performance are all detected, sarcopenia is considered severe [2].

A wide variety of tests and tools are available for the assessment of muscle in clinical practice, the selection of which may depend upon the availability of resources in the healthcare setting. The SARC-F questionnaire is a five-item simple clinical symptom index that is self-reported by patients as a screen for sarcopenia risk [3]. Measuring handgrip strength is also simple and inexpensive, but accurate measurement requires the use of a calibrated handheld dynamometer under well-defined test conditions with interpretive data from appropriate reference populations [4]. Based on data of two studies [5,6], normal reference values for the Spanish population have been reported.

On the other hand, measuring the quantity and quality of muscle mass has been positioned as a crucial aspect for the diagnosis of disease-related malnutrition (DRM). In this context, nutritional ultrasound that evaluates fat-free mass is an emerging cheap, portable, and non-invasive technique that quantifies muscle in malnutrition [7,8,9], with advantages over computed tomography (CT), magnetic resonance imaging (MRI), or dual photon X-ray absorptiometry (DXA) techniques that may be less accessible in clinical practice and involve a high healthcare cost, especially CT and MRI [8]. Bioelectrical impedance analysis (BIA) of muscle mass may be preferable to DXA, but validated prediction equations for specific populations are necessary [2]. In addition, DXA and BIA do have cut-off values for muscle quantity, but these methods do not provide indexes for muscle quality, whereas CT and MRI can measure both muscle quantity and quality, but clear cut-off points are still undefined [10].

Muscle ultrasound evaluates muscle volume and area, the length of the fascicles, and the angle of the muscle pennation in transverse and longitudinal positions, as well as subcutaneous fat [9]. However, standardization of methods and measures is still needed. In 2018, the SARCUS (SARCopenia through UltraSound) Working Group [10] reported a consensus proposition for anatomical landmarks of ultrasonographic muscle assessment, with recommendations for patient positioning, system settings, and components to be measured. The application of ultrasound to measure sarcopenia has been recently updated by the SARCUS group, including a detailed description of measuring points and muscle parameters for 39 muscles/muscle groups [11]. In a previous study of our group, standardization of the ultrasound measurement of quadriceps rectus femoris for use in clinical practice was described [9].

However, data on ultrasound cut-off values for predicting low muscle mass status are scarce. Sari et al. [12] reported cut-off values for the gastrocnemius medialis and rectus abdominis in patients with systemic sclerosis. Barotsis et al. [13] predicted sarcopenia from ultrasonographically measured muscle thickness of the vastus intermedius, rectus femoris, medial head of the gastrocnemius, and geniohyoid based on receiver operating characteristic (ROC) analysis. Fukumoto et al. [14] estimated cut-off values of the rectus femoris, vastus intermedius, gastrocnemius, and soleus muscles to detect low muscle mass for sarcopenia. Finally, Eşme et al. [15] reported cut-off values for the gastrocnemius, rectus femoris cross-sectional area, and external and internal oblique for predicting sarcopenia in patients with sarcoidosis.

This prospective multicenter cohort study was designed to assess the usefulness in clinical practice of nutritional ultrasound for the diagnosis of sarcopenia in patients at nutritional risk and to establish cut-off values of different ultrasound measures in patients at risk of sarcopenia and in those with probable, confirmed, and severe sarcopenia.

## 2. Materials and Methods

### 2.1. Design and Study Population

This was a nationwide, prospective, multicenter, cohort study (the DRECO study, “Disease-Related caloric-protein malnutrition EChOgraphy”) carried out at the Services of Endocrinology and Nutrition of public hospitals throughout Spain. The objectives of the study were to assess the contribution of ultrasound of the rectus femoris for diagnosing sarcopenia in hospitalized patients at risk of malnutrition, and to define cut-off values of ultrasound parameters for the identification of risk of sarcopenia and probable, confirmed, and severe sarcopenia.

Between March and December 2022, consecutive patients aged 18 to 85 years admitted to medical–surgical departments of the participating hospitals (excluding intensive care units [ICUs]) who were diagnosed of being at risk of malnutrition during the first week of hospital stay were eligible if informed consent had been obtained. Exclusion criteria were the presence of liver dysfunction (aminotransferase levels > 3 times the upper reference limit); chronic renal failure (glomerular filtration rate < 45 mL/min/1.73 m^2^); previous ICU stays during the index hospital admission; cancer patients with Eastern Cooperative Oncology Group (ECOG) performance status ≥ 3 points [16]; eating disorders; any musculoskeletal disease preventing unassisted walking ability; dementia, cognitive impairment, or any neurological/psychiatric condition that may interfere with the study procedures; a life expectancy of less than 6 months; and refusal to sign the informed consent form.

The study protocol was approved by the Ethics Committee for Clinical Research (CEIC) of the Health Council of the Andalusian Health Service (protocol code ALM-DRECO-2021-01, approval date 1 February 2022) and the individual Institutional Review Boards of the participating hospitals. Written informed consent was obtained from all patients. The study was conducted in accordance with the principles of the Declaration of Helsinki and registered in ClinicalTrials.gov (NCT05433831) https://clinicaltrials.gov/study/NCT05433831 (accessed on 14 May 2024).

### 2.2. Assessment of Malnutrition and the Risk of Sarcopenia

Screening for the risk of sarcopenia was assessed using the SARC-F questionnaire [17,18] and the malnutrition risk was assessed by the Malnutrition Universal Screening Tool (MUST) [19].

Risk of sarcopenia was defined in the presence of an SARC-F score ≥ 4.

### 2.3. Ultrasound Measurements

Ultrasound measurements of the unilateral (right side) rectus femoris were performed at each participating center by an experienced medical sonographer blinded to the clinical data and other results of nutritional assessment using a commercially available portable ultrasound system with a 4–10 cm linear tube (UProbe L6C Ultrasound Scanner, Guangzhou Sonostar Technologies Co., Ltd., Guangzhou, China). Abdominal and anterior thigh muscle measurements were performed with the patient lying supine with their knees extended and relaxed. A linear 7.5–10 kHz ultrasound probe was used. The acquisition site was located two-thirds of the way along the femur length, measured between the anterior superior iliac spine and the upper edge of the patella. The transducer was placed perpendicular to the long axis of the thigh with excessive use of contact gel and minimal pressure to avoid compression of the muscle. All parameters were taken as an average of three consecutive measurements in the dominant leg. We measured the transversal axis of the cross-sectional area (CSA) in cm^2^; the *X*-axis and *Y*-axis in mm, which corresponded to the linear measurement of the distance between the muscular limits of the rectus femoris (lateral and anteroposterior); the *X*-axis/*Y*-axis ratio; and the total fat tissue in mm. All US parameters were also normalized and divided by height squared (in cm^2^ for rectus femoris).

### 2.4. Study Variables

Other data recorded included sociodemographic and anthropometric characteristics, handgrip strength, bioimpedance analysis (BIA), the Timed Up and Go (TUG) test, and biochemical data. Handgrip strength was determined using the Jamar dynamometer (J A Preston Corporation, New York, NY, USA). The dominant hand was tested. Three measurements were taken, and the average was reported and compared with the published population reference data that were used as cut-off points [5]. Total body BIA (50 kHz frequency) (Akern EFG BIA 101 Anniversary) was used to determine phase angle (degrees), total body water (%), fat mass (kg), lean mass (kg), body cell mass (kg), and appendicular skeletal muscle mass (ASMM) (kg). The TUG test was used to assess functionality. A colored tape was marked 3 m away from an armless chair in which participants were sitting. Participants were asked to walk 3 m, turn around the marked tape, and return to the chair as fast as they could. A timer was set as soon as the patient stood up from the chair and was stopped when the patient was seated again. At least one practice trial was performed before the test. A TUG-score of ≥20 s was identified as a cut-off point for sarcopenia [2]. Biochemical variables included serum levels of albumin (g/dL), prealbumin (g/dL), C-reactive protein (CRP) (mg/L), and the CRP/prealbumin ratio.

### 2.5. Categories of Sarcopenia

The presence of risk of sarcopenia was defined by the identification of an SARC-F score ≥ 4; probable sarcopenia was defined by an SARC-F score ≥ 4 and low handgrip strength based on cut-off reference values (10th percentile) for the Spanish population [5]. In all patients, sarcopenia assessment was carried out according to EWGSOP2 criteria to detect confirmed sarcopenia as criteria of probable sarcopenia plus abnormal ASMM on BIA (<7.0 kg/m^2^ for men and <5.5 kg/m^2^ for women) [2] and severe sarcopenia as criteria of confirmed sarcopenia plus TUG ≥ 20 s [2].

### 2.6. Outcomes

The primary outcome of this study was to assess the usefulness of ultrasound of the rectus femoris for detecting sarcopenia in hospitalized patients at risk of malnutrition. The secondary outcome was to define cut-off values of the different ultrasound measures for the diagnosis of risk of sarcopenia, probable sarcopenia, confirmed sarcopenia, and severe sarcopenia.

### 2.7. Statistical Analysis

For the purpose of this study, the sample was distributed by quotas to cover 50% men and 50% women and stratified by 10-year age ranges. It was estimated that a large sample of 1000 patients would be adequate to assess the outcomes of the study. The inclusion of at least 40 patients per center was expected from about 20–25 hospitals. Patients were admitted to the Services of Endocrinology and Nutrition, in which screening for disease-related malnutrition is routinely performed, and referred to a nutritional support team to complete the nutritional assessment and treatment.

Categorical variables are expressed as frequencies and percentage, and continuous variables as mean and standard deviation (±SD). The chi-square test or Fisher’s exact test were used for the comparison of qualitative variables, and Student’s *t* test, two-way analysis of variance (ANOVA), the Mann–Whitney U test, or the Kruskal–Wallis test for the comparison of quantitative variables according to conditions of application. Bonferroni correction was applied as a multiple comparison procedure. The correlation between ultrasound variables (CSA, *X*-axis, *Y*-axis) and mean handgrip strength, BIA (body cell mass), and TUG was assessed with the Spearman rank-order correlation coefficient (rho). Correlations of 0–0.19 were regarded as very weak, 0.2–0.39 as weak, 0.40–0.59 as moderate, 0.6–0.79 as strong, and 0.8–1 as very strong. Receiver operating characteristic (ROC) and area under the curve (AUC) analyses were used to determine the optimal diagnostic sensitivity, specificity, and predictive values of cut-off points of the ultrasound measures for the detection of risk of sarcopenia and probable, confirmed, and severe sarcopenia. The cut-off points were determined by the AUC method that showed the best specificity and sensitivity values for the test in question, as well as the Youden index (sensitivity + specificity − 1). Analyses were performed for the overall study population as well as separately for men and women. Statistical significance was set at *p* < 0.05. Statistical Analysis System (SAS) version 9.4 was used for data analysis.

## 3. Results

### 3.1. General Characteristics of Patients

During the study period, a total of 1000 hospitalized patients were screened for risk of malnutrition; 9 of them refused to participate in the study after inclusion, so 991 patients were finally included in the study (58.9% men and 41.1% women). The mean age was 58.5 ± 16.5 years, mean weight 63.6 ± 14.8 kg, and mean body mass index (BMI) 22.9 ± 4.8 kg/m^2^. Sociodemographic and anthropometric characteristics, risk of malnutrition, and results of dynamometry, BIA, TUG, and biochemical variables in all patients as well as in men and women are shown in Table 1.

There were statistically significant differences between men and women in most study variables, except for the risk of malnutrition, biochemical variables, and the percentages of patients with normal or abnormal handgrip strength and ASMM when the corresponding cut-off points recommended by the EWGSOP2 [2] were applied. Women compared with men showed significantly lower values of BMI, mean handgrip strength, all BIA parameters except for fat mass, and all ultrasound measures except for total fat tissue and preperitoneal and total fat on abdominal ultrasound examination. The percentage of women with an abnormal TUG test was significantly higher than that of men (Table 1).

### 3.2. Prevalence of Sarcopenia

As shown in Table 2, most patients (62.8%) were not at risk of sarcopenia and did not fulfill the criteria for sarcopenia. Risk of sarcopenia was identified in 9.6% of patients and probable sarcopenia in 14.0%. Confirmed sarcopenia was found in 9.7% of patients and severe sarcopenia in 3.9%. There were statistically significant differences (*p* < 0.0001) in the distribution of categories of sarcopenia between men and women, with higher percentages of absence of sarcopenia and confirmed and severe sarcopenia among men, whereas the risk of sarcopenia and probable sarcopenia was more common among women (Figure 1).

### 3.3. Correlation between Ultrasound Variables, Handgrip Strength, BIA, and TUG

The CSA of the rectus femoris showed a significantly positive correlation the *X*-axis, *Y*-axis, body cell mass of BIA, and handgrip strength, and a significant negative correlation with TUG. The *X*-axis and *Y*-axis showed the same pattern than the CSA, with significant positive correlations with body cell mass of BIA and handgrip strength, and negative correlations with TUG. Body cell mass of BIA and handgrip strength correlated significantly with ultrasound variables but showed a negative correlation with TUG. In general, correlations ranged between moderate and strong, but in the case of TUG, correlations were mostly weak (Table 3).

### 3.4. Ultrasound Cut-Off Points for Detecting Sarcopenia

Cut-off values of the main ultrasound measures in the groups of patients categorized by risk of sarcopenia, probable sarcopenia, confirmed sarcopenia, and severe sarcopenia as well as according to sex are shown in Table 4. In general, the cut-off values were similar within each category of sarcopenia, ranging from 2.40 cm^2^ to 3.66 cm^2^ for CSA, 32.57 mm to 40.21 mm for the *X*-axis, and 7.85 mm to 10.4 mm for the *Y*-axis. In general, these cut-off values were associated with high sensitivities for all ultrasound measures, particularly for the categories of confirmed and severe sarcopenia, with male patients also showing better sensitivities compared with females. However, specificities and positive predictive values were low, but negative predictive values were consistently high. The most favorable cut-off value was 8.65 mm for the *Y*-axis for men with severe sarcopenia, with an AUC of 0.801, sensitivity of 80.8%, and specificity of 77.3%, followed by 3.48 cm^2^ for the CSA in men with confirmed sarcopenia, with an AUC of 0.777, sensitivity of 81.4%, and specificity of 66.9%.

## 4. Discussion

It is well known that sarcopenia is one of the most important health problems in elderly people with a high rate of adverse outcomes. Data of a systematic review and meta-analysis of 35 articles with a total of 58,404 individuals revealed an overall prevalence of 10% in both men and women, with a substantial proportion of old people having sarcopenia, even in healthy populations [20]. Sarcopenia has been associated with an increased risk of mortality, falls, fractures, and poor quality of life [21], so timely detection can be effective in reducing the burden of disease. In this respect, ultrasound provides a safe, cost-effective, and rapid means of assessing the musculoskeletal system [22] and is very promising in geriatric practice in the context of sarcopenia [23]. The present real clinical practice study in shows that in a large population of inpatients undergoing routine screening for the risk of malnutrition, ultrasound examination of the rectus femoris was a feasible technique for detecting sarcopenia, particularly in cases of confirmed and severe sarcopenia defined by a combination of SARC-F score, handgrip strength, ASMM on BIA, and results of TUG. It should be noted that definitions of these variables were based on standard interpretation of the SARC-F questionnaire (≥4 points) and the use of reference values for handgrip strength using a Jamar dynamometer already reported in a Spanish population by gender and age groups [5] and cut-points of ASMM and TUG proposed by the EWGSOP2 group for the diagnosis of sarcopenia [2]. In fact, the strict definitions of the categories of sarcopenia (at risk, probable, confirmed, and severe) for which ultrasound cut-off values of the rectus femoris have been estimated are a strength of this study and an important contribution of the present findings.

There are few studies on thigh muscle evaluation by ultrasound of the rectus femoris in the diagnosis of sarcopenia [13,14,15], but a direct comparison with our findings cannot be established due to methodological differences in acquisition points and the ultrasound parameters considered. Ultrasound measurements of abdominal and calf muscle thickness was found to be a useful screening method in predicting low-muscle-mass status in patients with systemic sclerosis, with a high sensitivity (92.3%) for both the gastrocnemius medialis and rectus abdominis and negative predictive value (97.9% and 97.6%, respectively) [12]. In this study, however, ultrasound assessment of the rectus femoris was not performed. In another prospective study of 94 individuals with a mean age of 75.6 years referred for sarcopenia screening to a rehabilitation department of a university hospital in Patras, Greece [13], thickness of the rectus femoris was measured between its deep and superficial fascia. It was found that the likelihood of sarcopenia was 11.9 and 6.9 times greater for transverse and longitudinal section thickness lower than the cut-off points of 1.54 cm and 1.59 cm, respectively, for which sensitivities of 68.8% and 81.3% and specificities of 65.4% and 51.3% were reported [13]. These data, however, are difficult to compare with our study as the acquisition points were not described. In a cross-sectional study of 204 community-dwelling older adults (mean age 75.4 years) and 59 younger adults (mean age 22.3 years), lower limb muscle thickness was evaluated to assess sarcopenia [14]. The cut-off point of rectus femoris muscular thickness based on 2 SD below the young adults was 1.85 cm for males and 1.42 cm for females, corresponding to a prevalence of low muscular mass of 69.4% and 36.7%, respectively. In this study, the muscular thickness of the rectus femoris was defined as the distance between the superficial and deep fascia of the muscle. Finally, in a study of 40 patients with a mean age of 53.2 years, a cut-off value of the rectus femoris cross-sectional area of 5.65 mm^2^ showed a sensitivity of 76% and a specificity of 69% for predicting sarcopenia [15]. In our study, CSA showed cut-off points ranging between 3.37 and 3.66 cm^2^, with sensitivities of 58.5% for predicting the risk of sarcopenia and 64.4%, 81%, and 78.1% for probable, confirmed, and severe sarcopenia, respectively.

An interesting aspect of our study was the analysis of the distribution of the study variables by sex, with values in general being higher in male patients than in female patients. The assessment of differences in nutritional-related variables between men and women provides valuable information at the time of targeting nutritional interventions. In a systematic review and meta-analysis of 107 RCTs, a greater proportion of gender-targeted interventions than gender-neutral studies were effective in improving nutrition [24]. In relation to ultrasound cut-off values for the evaluation of sarcopenia, men show higher cut-off points than women in practically all categories of sarcopenia, a fact that should be taken into consideration in practice. However, differences in cut-off values of patients stratified by age were not evaluated. On the other hand, as may be expected, there were statistically significant correlations between ultrasound variables and handgrip strength, BIA, and TUG, which is consistent with data reported in previous studies [14,15].

The field of US muscle assessment is clearly growing, with more research groups using this technique to give more hands-on information on the muscles described. However, a clear standardization remains absent. A large number of variables can influence the use of US for the determination of sarcopenia. The first factor is the location of the muscle that we can measure; a multitude of areas, up to 39 upper extremity muscles (upper arm, lower arm, and hand), lower extremity muscles (upper leg, lower leg, foot), and head and neck muscles, have been evaluated in the literature [11]. We decided to measure the rectus femoris [25,26,27,28,29] as a well-known muscle with previous clinical studies. It is one of the most evaluated muscles in the literature and very accessible to an untrained observer, and therefore, each one must have sarcopenia cut-off points and different parameters (*x*-axis, *y*-axis, circumference, area, fascicle length, echo-intensity pennation angle and so on). In our present study, we report the cut-off points for sarcopenia in this specific muscle. Second, whereas a resting period of a minimum of 30 min was previously proposed, new data show that when changing from a standing to a supine position, after 5 min, a normalization of measurements can occur. In our protocol, the US image was captured in the supine position. Finally, some muscles can easily be delineated through the use of specific anatomical landmarks, but others will still require an ultrasonographic visualization before exact measuring points can be identified, for example, suprahyoid musculature of the neck or the flexor hallucis brevis in the foot [11]. A recent revision of Niels et al. [30] indicated that ultrasound of the rectus femoris muscle to diagnose sarcopenia has been shown to be a promising method in multiple clinical populations and it is necessary to implement protocols in clinical practice [31], like our present study.

Several limitations should be noted when interpreting this study. The results may not be generalizable to other muscle groups, as only the rectus femoris was assessed. However, this location of muscle is easily accessible for ultrasound in the supine position and has an excellent association with whole body muscle mass [31]. Although there are different muscle structures that can be evaluated, many studies focus on the rectus femoris or combinations of various muscle groups involving large muscle bundles with functional importance to patients in terms of gait. Measurement of the rectus femoris of the quadriceps is one of the most referenced measurements due to its correlation with strength and tests of execution or functional performance [25,26,27,28,29,32]. The data from our work can be extrapolated only to patients at potential risk of malnutrition when hospitalized and older than 18 years. Our data cannot be generalized to ICU patients, considering the design and inclusion criteria of our protocol. The data may vary depending on the image acquisition equipment as well as the protocol used for the acquisition of these ultrasound images; thus, the DRECO study protocol has been recently published [33]. Inter- and intra-observer variability may be a confounding factor in our results that should be considered in future studies. Finally, the absence of recording physical activity may be a limitation in the interpretation of the results.

However, the large sample size and the assessment of global cut-off values of ultrasound measures of the rectus femoris, as well as those for men and women, are important strengths and differential features of this study. Also, estimates of cut-off values according to the categories of sarcopenia are relevant scientific contributions of this study.

## 5. Conclusions

In a large population of patients admitted to the medical–surgical departments of public hospitals throughout Spain who were routinely screened for risk of malnutrition using validated instruments, 9.6% were at risk of sarcopenia, 14% had probable sarcopenia, and 9.7% had confirmed sarcopenia. Severe sarcopenia was detected in almost 4%. Based on these categories of sarcopenia, cut-off values for the better sensitivities and specificities of different ultrasound measures of the rectus femoris are established for the global study population as well as for male and female patients. Ultrasound of the rectus femoris can be used for the prediction of sarcopenia. The findings of the present clinical study are useful to integrate nutritional ultrasound in real clinical practice.

## Figures and Tables

**Figure 1 nutrients-16-01552-f001:**
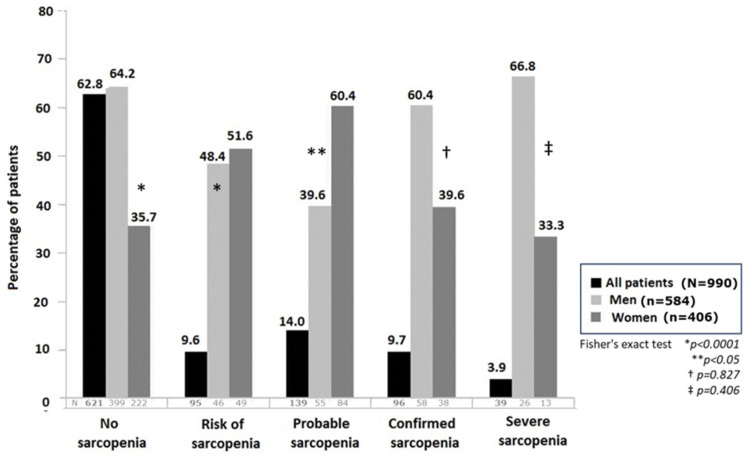
Percentage of patients in the different categories of sarcopenia for the overall study population and distributed by sex.

**Table 1 nutrients-16-01552-t001:** General characteristics of patients and distribution of study variables by sex.

Variables	All Patients	Men	Women	Difference	*p*
	(*n* = 991)	(*n* = 585)	(*n* = 406)	(*n* = 991)	
Age, years, mean ± SD	58.5 ± 16.5	58.9 ± 16.5	57.8 ± 16.3	−1.1 ± 1.1	0.33
Weight, kg, mean ± SD	63.7 ± 14.8	68.5 ± 14.2	56.7 ± 12.9	−11.8 ± 0.9	<0.0001
BMI, kg/m^2^, mean ± SD	22.9 ± 4.8	23.4 ± 4.7	22.3 ± 4.9	−1.1 ± 0.3	0.0004
Handgrip strength, kg, mean ± SD (*n* = 963)	25.0 ± 10.8	30.0 ± 10.2	17.8 ± 6.8	−12.2 ± 0.5	<0.0001
EWGSOP2 cut-off (men 27 kg, women 16 kg) (*n* = 963)					
Normal, *n* (%)	321 (33.3)	188 (33.1)	133 (33.7)	−55 (0.60)	0.889
Abnormal, *n* (%)	642 (66.7)	380 (66.9)	262 (66.3)	−118 (0.60)	
**BIA, mean ± SD**					
Phase angle, degrees, (*n* = 907)	5.02 ± 1.11	5.20 ± 1.17	4.76 ± 0.96	−0.44 ± 0.1	<0.0001
Total body water, % (*n* = 939)	73.53 ± 6.14	74.05 ± 5.85	72.75 ± 6.48	−1.3 ± 0.4	0.001
Fat mass, kg (*n* = 958)	15.13 ± 8.40	14.63 ± 8.17	15.85 ± 8.67	1.22 ± 0.5	0.027
Lean mass, kg (*n* = 968)	48.06 ± 10.09	53.36 ± 8.90	40.52 ± 6.10	−12.84 ± 0.5	<0.0001
Body cell mass, kg (*n* = 934)	23.46 ± 6.40	26.25 ± 6.14	19.46 ± 4.28	−6.79 ± 0.3	<0.0001
Appendicular skeletal muscle mass, kg/m^2^ (*n* = 937)	6.33 ± 1.63	6.75 ± 1.71	5.72 ± 1.27	−1.03 ± 0.1	<0.0001
EWGSOP2 cut-off (men 7 kg/m^2^, women 5.5 kg/m^2^) (*n* = 937)					
Normal, *n* (%)	474 (50.6)	265 (47.4)	209 (55.3)	−56 (7.9)	0.019
Abnormal, *n* (%)	463 (49.4)	294 (52.6)	169 (44.7)	−125 (−7.9)	
**Ultrasound rectus femoris, mean ± SD**					
Cross-sectional area, cm^2^ (*n* = 869)	3.80 ± 1.37	4.09 ± 1.42	3.33 ± 1.13	−0.76 ± 0.1	<0.0001
*X*-axis, mm, (*n* = 979)	37.16 ± 5.87	38.65 ± 5.73	34.99 ± 5.39	−3.66 ± 0.4	<0.0001
*Y*-axis, mm (*n* = 981)	10.45 ± 3.54	11.10 ± 3.80	9.51 ± 2.89	−1.59 ± 0.2	<0.0001
*X*-axis/*Y*-axis ratio, mm (*n* = 979)	3.93 ± 1.35	3.88 ± 1.42	4.0 ± 1.26	0.12 ± 0.1	0.18
Total fat tissue, mm (*n* = 940)	7.09 ± 4.73	5.44 ± 3.38	9.41 ± 5.35	3.97 ± 0.3	<0.0001
TUG, s, mean ± SD (*n* = 829)	13.65 ± 7.70	12.53 ± 6.64	15.21 ± 8.73	2.68 ± 0.5	<0.0001
EWGSOP2 cut-off ≥ 20 s in men and women, (*n* = 829)					
Normal, *n* (%)	696 (84.0)	426 (88.4)	270 (77.8)	−156 (−10.6)	0.0005
Abnormal, *n* (%)	133 (16.0)	56 (11.6)	77 (22.2)	21 (10.6)	
**Biochemical data, mean ± SD**					
Albumin, g/dL (*n* = 925)	3.45 ± 0.76	3.45 ± 0.73	3.45 ± 0.81	0 ± 0.1	0.977
Prealbumin, mg/dL (*n* = 677)	17.89 ± 8.22	17.77 ± 8.47	18.07 ± 7.86	0.30 ± 0.5	0.638
C-reactive protein (CRP), mg/L (*n* = 905)	45.56 ± 65.97	48.0 ± 63.8	42.1 ± 68.9	−5.9 ± 4.3	0.185
CPR/prealbumin ratio (*n* = 659)	5.15 ± 12.62	5.90 ± 13.14	4.09 ± 11.78	−1.81 ± 0.8	0.07

BMI: body mass index. CRP: C-reactive protein. SD: standard deviation; GLIM: Global Leadership Initiative on Malnutrition; SGA: Subjective Global Assessment; BIA: bioimpedance analysis; EWGSOP2: European Working Group on Sarcopenia in Older People; TUG: Timed Up and Go; low handgrip strength based on cut-off reference values (10th percentile) for the Spanish population [5].

**Table 2 nutrients-16-01552-t002:** Categories of sarcopenia and distribution by sex.

Categories	All Patients (*n* = 990)	Men (*n* = 584)	Women (*n* = 406)	Difference (*n* = 990)
Sarcopenia, *n* (%)				
Absence	621 (62.8)	399 (68.3)	222 (54.7)	−177 (−13.6)
At risk	95 (9.6)	46 (7.9)	49 (12.1)	3 (4.2)
Probable	139 (14.0)	55 (9.4)	84 (20.7)	29 (11.3)
Confirmed	96 (9.7)	58 (9.9)	38 (9.4)	−20 (−0.5)
Severe	39 (3.9)	26 (4.5)	13 (3.2)	−13 (−1.3)

Risk for sarcopenia: in all patients, sarcopenia assessment was carried out according to EWGSOP2 criteria to detect confirmed sarcopenia as criteria of probable sarcopenia plus abnormal ASMM on BIA (<7.0 kg/m^2^ for men and <5.5 kg/m^2^ for women) and severe sarcopenia as criteria of confirmed sarcopenia plus TUG ≥ 20 s.

**Table 3 nutrients-16-01552-t003:** Correlations between the study variables.

Variables	CSA cm^2^	*X*-Axis mm	*Y*-Axis mm	Handgrip Strength, kg	BIA, Body Cell Mass, kg	TUG, s
CSA, cm^2^	-	*n* = 867 rho = 0.624 *p* < 0.001	*n* = 869 rho = 0.788 *p* < 0.001	*n* = 850 rho = 0.426 *p* < 0.001	*n* = 822 rho = 0.519 *p* < 0.001	*n* = 738 rho = −0.290 *p* < 0.001
*X*-axis, mm	*n* = 867 rho = 0.624 *p* < 0.001	-	*n* = 979 rho = 0.393 *p* < 0.001	*n* = 955 rho = 0.411 *p* < 0.001	*n* = 924 rho = 0.368 *p* < 0.001	*n* = 822 rho = −0.246 *p* < 0.001
*Y*-axis, mm	*n* = 869 rho = 0.788 *p* < 0.001	*n* = 979 rho = 0.393 *p* < 0.001	-	*n* = 957 rho = 0.391 *p* < 0.001	*n* = 926 rho = 0.548 *p* < 0.001	*n* = 823 rho = −0.340 *p* < 0.001
Handgrip strength, kg	*n* = 850 rho = 0.425 *p* < 0.001	*n* = 955 rho = 0.411 *p* < 0.001	*n* = 957 rho = 0.391 *p* < 0.001	-	*n* = 912 rho = 0.633 *p* < 0.001	*n* = 815 rho = −0.466 *p* < 0.001
BIA, body cell mass, kg	*n* = 822 rho = 0.519 *p* < 0.001	*n* = 924 rho = 0.368 *p* < 0.001	*n* = 926 rho = 0.548 *p* < 0.001	*n* = 912 rho = 0.633 *p* < 0.001	-	*n* = 786 rho = −0.300 *p* < 0.001
TUG, s	*n* = 738 rho = −0.290 *p* < 0.001	*n* = 822 rho = −0.242 *p* < 0.001	*n* = 823 rho = −0.340 *p* < 0.001	*n* = 815 rho = −0.466 *p* < 0.001	*n* = 786 rho = −0.300 *p* < 0.001	-

CSA: cross-sectional area by ultrasound of rectus femoris; BIA: bioimpedance analysis; TUG: Timed Up and Go; *X* axis by ultrasound of rectus femoris; *Y*-axis by ultrasound of rectus femoris; *X*/*Y* axis ratio by ultrasound of rectus femoris; rho: Spearman’s correlation coefficient; *n* = number of patients.

**Table 4 nutrients-16-01552-t004:** Cut-off points of ultrasound variables of the rectus femoris for detecting sarcopenia in all study patients and distributed by sex.

Variables	Sarcopenia Category	Study Patients	Cut-Off Value	AUC	Sensitivity %	Specificity %	Predictive Values	
							Positive %	Negative %
Cross-sectional area (CSA), cm^2^	Risk of sarcopenia	All patients	3.37	0.629	58.5	61.5	45.8	72.7
		Men	3.48	0.647	56.4	66.6	42.4	77.8
		Women	2.97	0.556	50.0	62.5	51.7	60.8
	Probable sarcopenia	All patients	3.37	0.634	64.4	59.1	28.7	86.6
		Men	3.48	0.700	66.7	66.9	35.9	87.8
		Women	3.37	0.548	70.0	41.5	20.9	86.2
	Confirmed sarcopenia	All patients	3.66	0.680	81.0	49.5	20.0	94.3
		Men	3.48	0.777	81.4	66.4	26.9	85.9
		Women	2.4	0.483	89.1	16.3	14.8	90.2
	Severe sarcopenia	All patients	3.41	0.669	78.1	55.3	6.4	98.5
		Men	3.41	0.818	95.2	66.6	10.8	99.7
		Women	3.12	0.597	72.7	49.7	4.8	98.1
*X*-axis, mm	Risk of sarcopenia	All patients	37.37	0.583	58.3	56.0	44.3	69.1
		Men	40.1	0.579	68.6	45.8	37.2	75.7
		Women	37.41	0.534	72.5	35.0	48.3	60.3
	Probable sarcopenia	All patients	33.55	0.610	37.6	79.3	34.5	80.4
		Men	40.21	0.634	77.4	46.0	30.7	86.8
		Women	32.57	0.620	51.2	73.7	34.1	84.9
	Confirmed sarcopenia	All patients	38.3	0.579	73.3	46.7	18.3	91.5
		Men	38.3	0.687	76.2	59.8	24.6	93.6
		Women	34.41	0.584	74.5	43.4	16.4	91.9
	Severe sarcopenia	All patients	38.3	0.613	76.9	45.4	5.7	97.8
		Men	37.82	0.725	76.9	62.4	9.2	98.2
		Women	37.69	0.579	53.8	67.9	5.8	97.8
*Y*-axis, mm	Risk of sarcopenia	All patients	9.59	0.628	56.9	63.5	48.3	71.2
		Men	9.66	0.652	55.7	70.2	46.6	77.2
		Women	8.57	0.563	48.9	65.1	53.9	60.4
	Probable sarcopenia	All patients	9.59	0.645	62.4	61.2	31.8	84.7
		Men	9.66	0.691	64.2	70.0	39.8	86.4
		Women	7.85	0.583	44.0	73.7	30.8	83.2
	Confirmed sarcopenia	All patients	9.66	0.686	71.9	59.4	22.3	92.8
		Men	9.66	0.775	78.6	69.7	30.8	94.9
		Women	10.4	0.534	74.5	35.4	14.7	90.3
	Severe sarcopenia	All patients	8.77	0.716	74.4	67.6	9.0	98.4
		Men	8.65	0.801	80.8	77.3	14.9	98.8
		Women	8.77	0.558	61.5	56.0	4.6	97.7
*X*/*Y* axis ratio	Risk of sarcopenia	All patients	5.19	0.598	89.9	25.3	60.0	66.7
		Men	4.63	0.624	35.7	83.8	50.7	73.5
		Women	4.95	0.552	24.7	86.6	60.81	57.8
	Probable sarcopenia	All patients	4.63	0.598	37.1	79.9	35.0	81.3
		Men	4.64	0.638	40.1	83.1	42.3	81.7
		Women	4.95	0.533	27.4	83.8	31.1	81.2
	Confirmed sarcopenia	All patients	4.19	0.661	60.0	68.3	23.5	91.2
		Men	4.66	0.708	52.4	84.0	36.1	91.1
		Women	4.16	0.582	62.7	61.0	19.4	91.6
	Severe sarcopenia	All patients	4.19	0.666	66.7	66.7	79.5	97.9
		Men	4.67	0.577	57.7	82.2	13.8	97.5
		Women	4.26	0.602	69.2	63.0	6.08	98.3

Risk for sarcopenia: in all patients, sarcopenia assessment was carried out according to EWGSOP2 criteria to detect confirmed sarcopenia as criteria of probable sarcopenia plus abnormal ASMM on BIA (<7.0 kg/m^2^ for men and <5.5 kg/m^2^ for women) and severe sarcopenia as criteria of confirmed sarcopenia plus TUG ≥ 20 s.

## Data Availability

The original contributions presented in the study are included in the article, further inquiries can be directed to the corresponding author due to privacy reasons.

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
