# Peer review of "Ultrasound Cut-Off Values for Rectus Femoris for Detecting Sarcopenia in Patients with Nutritional Risk"

_nutrients, 2024, doi:10.3390/nu16111552_

Round 1
Reviewer 1 Report
Comments and Suggestions for Authors
The manuscript "Ultrasound cut-off values for quadriceps rectus femoris for detecting sarcopenia in patients with nutritional risk" presents the results of a nationwide, prospective, multicenter, cohort study that aimed to define cut-off values for different sarcopenia-related measures in patients at risk of malnutrition. Among the strengths of this study, the large sample size (n= 991) and the variety of assessments are particularly relevant.
I only have some questions/suggestions:
- Are the authors able to eventually report the distribution of the main reasons why participants were hospitalized? It might be helpful to support the generalization of the findings and further considerations
- Was physical activity level also collected, for example with the IPAQ or other questionnaires? It might interesting to see the association between physical activity and the assessed outcomes.
- For uniformity of reporting, I suggest always using the same number of decimals of the p-value (e.g., two or three decimals); exceptions can be made for values < 0.0001 < 0.00001, and so on. For example, "EWGSOP2 cut-off (men 27 kg, women 16 kg) (n = 963)..." in Table 1, you can only report 0.889 as p-value.
- Since the very interesting study, I suggest including effect sizes of the statistical tests as providing only p values allows a partial interpretation of the reported differences.
- If I am not wrong, Table 2 and Figure 1 present the same data. I recommend not duplicating the data and choosing only one of the two. Also for Figure 1, I suggest including more details in its caption as recommended, such as sample size and what statistical tests were performed. Finally, you can still present only 3 decimals for the p-values.
- The authors state that a limitation of the study was the US assessment of only the rectus femoris muscle. I suggest the authors clearly state why they chose to perform the measurements on that muscle, and report what other muscles could be assessed in different settings, including the vastus lateralis or the forearm muscles for the upper limb (Ticinesi et al., 2018; Di Lenarda et al., 2024; Abe et al., 2014).
Author Response
Dear Reviewer,
Thank you very much for taking the time to review this manuscript. Please find the detailed responses below and the corresponding revisions/corrections highlighted/in track changes in the re-submitted files.
Best,
Dr. Daniel de Luis
The manuscript "Ultrasound cut-off values for quadriceps rectus femoris for detecting sarcopenia in patients with nutritional risk" presents the results of a nationwide, prospective, multicenter, cohort study that aimed to define cut-off values for different sarcopenia-related measures in patients at risk of malnutrition. Among the strengths of this study, the large sample size (n= 991) and the variety of assessments are particularly relevant.
I only have some questions/suggestions:
- Are the authors able to eventually report the distribution of the main reasons why participants were hospitalized? It might be helpful to support the generalization of the findings and further considerations
- We appreciate the reviewer's comments. The scientific committee decided to carry out the study on patients admitted with malnutrition related to the disease, with the purpose of homogenizing the sample and having similar technology to perform BIA, US and the rest of the tests. This sentence has been included in limitation section (The data from our work can be extrapolated only to patients at potential risk of malnutrition hospitalized and older than 18 years)
- Was physical activity level also collected, for example with the IPAQ or other questionnaires? It might interesting to see the association between physical activity and the assessed outcomes.
- Unfortunately, we have not included physical activity. For this reason, we have included this situation as a limitation to our study. "Finally, the absence of recording physical activity may be a limitation in the interpretation of the results."
- For uniformity of reporting, I suggest always using the same number of decimals of the p-value (e.g., two or three decimals); exceptions can be made for values < 0.0001 < 0.00001, and so on. For example, "EWGSOP2 cut-off (men 27 kg, women 16 kg) (n = 963)..." in Table 1, you can only report 0.889 as p-value.
- We have revised p-values (3 decimals)
- Since the very interesting study, I suggest including effect sizes of the statistical tests as providing only p values allows a partial interpretation of the reported differences.
- The effect sizes of Table 1 and 2 statistical tests have properly been included.
- If I am not wrong, Table 2 and Figure 1 present the same data. I recommend not duplicating the data and choosing only one of the two. Also for Figure 1, I suggest including more details in its caption as recommended, such as sample size and what statistical tests were performed. Finally, you can still present only 3 decimals for the p-values.
- Sample sizes and statistical test have properly been included in Figure 1. P-values have been checked to have only 3 decimals.
- The authors state that a limitation of the study was the US assessment of only the rectus femoris muscle. I suggest the authors clearly state why they chose to perform the measurements on that muscle, and report what other muscles could be assessed in different settings, including the vastus lateralis or the forearm muscles for the upper limb (Ticinesi et al., 2018; Di Lenarda et al., 2024; Abe et al., 2014).
- We have included this paragraph in order why we chose rectus femoris and 4 references with different findings with rectus femoris US
“Although there are different muscle structures that can be evaluated, many of the studies focus on the rectus femoris or on combinations of various muscle groups involving large muscle bundles with functional importance to the patient in terms of gait. Measurement of the rectus femoris of the quadriceps is one of the most referenced measurements due to its correlation with strength and tests of execution or functional performance” (26-31)
- Primo D., Izaola O., Gómez JJL, de Luis D. Correlation of the Phase Angle with Muscle Ultrasound and Quality of Life in Obese Females. Dis Markers. 2022 9:7165126
- López-Gómez JJ, Izaola-Jauregui O, Almansa-Ruiz L, Jiménez-Sahagún R, Primo-Martín D, Pedraza-Hueso M, Ramos-Bachiller B, González-Gutiérrez J, De Luis-Román D. Use of Muscle Ultrasonography in Morphofunctional Assessment of Amyotrophic Lateral Sclerosis (ALS) Nutrients 2024, 16, 1021.
- de Luis DA, Lopez Gomez JJ. Morphofunctional Nutritional assessment in clinical Practice: A new approach to assessing Nutritional Status. Nutrients 2023;15:4300-4303
- López-Gómez JJ, García-Beneitez D, Jiménez-Sahagún R, Izaola-Jauregui O, Primo-Martín D, Ramos-Bachiller B, Gómez-Hoyos E, Delgado-García E, Pérez-López P, De Luis-Román DA. Nutritional Ultrasonography, a Method to Evaluate Muscle Mass and Quality in Morphofunctional Assessment of Disease Related Malnutrition. Nutrients 2023, 15, 3923
- Lopez JJ, Gutierrez C, Izaola O, Primo D, Gomez E, Jimenez R, de Luis D. Real World practice study of the effect of a specific oral nutritional supplement for diabetes mellitus on the morphofuncitional assessment and proteinenergy requirements. Nutrients 2022;14, 4802

Reviewer 2 Report
Comments and Suggestions for Authors
Dear authors, congratulations for the nice study!
Abstract
quadriceps rectus femoris According to Terminologia anatomica 2020 the muscle is rectus femoris (being part of the quadriceps femoris muscle) If the authors consider, that the readers would easier localise rectus femoris to anterior thigh if they named it "quadriceps rectus femoris " I do not insist, although according to Terminologia anatomica terminus Quadriceps rectus femoris is not correct!
The comment applies also to the main text
Abbreviations in the abstract should be all explained, although they are explained in the main text
line 333 instead of standardized normalised, divided…
line 350 The CSA… to be added "of rectus femoris muscle"
In general captions to tables: all abbreviations should be explained as tables should be readable separate from the main text
Author Response
Dear Reviewer,
Thank you very much for taking the time to review this manuscript. Please find the detailed responses below and the corresponding revisions/corrections highlighted/in track changes in the re-submitted files.
Best,
Dr. Daniel de Luis
Abstract
quadriceps rectus femoris According to Terminologia anatomica 2020 the muscle is rectus femoris (being part of the quadriceps femoris muscle) If the authors consider, that the readers would easier localise rectus femoris to anterior thigh if they named it "quadriceps rectus femoris " I do not insist, although according to Terminologia anatomica terminus Quadriceps rectus femoris is not correct! The comment applies also to the main text
- We have modified the term throughout the work and have included the terminus “rectus femoris”
Abbreviations in the abstract should be all explained, although they are explained in the main text
- We have included abbreviations
line 333 instead of standardized normalised, divided…
- It has been corrected
line 350 The CSA… to be added "of rectus femoris muscle"
- It has been corrected
In general captions to tables: all abbreviations should be explained as tables should be readable separate from the main text
- All tables have been revised

Round 2
Reviewer 1 Report
Comments and Suggestions for Authors
I would like to thank the authors for considering my previous recommendations and for providing most of the suggested amendments. I feel that the manuscript has greatly improved.
I am still wondering about the importance of discussing previous literature and observing other muscles as markers of sarcopenia or muscle loss, as this would help to consider these findings in relationship with other findings and how depending on the muscle and technique of choice, parallels and differences can emerge.